# A Differential Equation for Modeling Nesterov's Accelerated Gradient Method: Theory and Insights

**Weijie Su**[1]     **Stephen Boyd**[2]     **Emmanuel J. Candès**[1,3]

[1]Department of Statistics, Stanford University, Stanford, CA 94305
[2]Department of Electrical Engineering, Stanford University, Stanford, CA 94305
[3]Department of Mathematics, Stanford University, Stanford, CA 94305
{wjsu, boyd, candes}@stanford.edu

## Abstract

We derive a second-order ordinary differential equation (ODE), which is the limit of Nesterov's accelerated gradient method. This ODE exhibits approximate equivalence to Nesterov's scheme and thus can serve as a tool for analysis. We show that the continuous time ODE allows for a better understanding of Nesterov's scheme. As a byproduct, we obtain a family of schemes with similar convergence rates. The ODE interpretation also suggests restarting Nesterov's scheme leading to an algorithm, which can be rigorously proven to converge at a linear rate whenever the objective is strongly convex.

## 1 Introduction

As data sets and problems are ever increasing in size, accelerating first-order methods is both of practical and theoretical interest. Perhaps the earliest first-order method for minimizing a convex function $f$ is the gradient method, which dates back to Euler and Lagrange. Thirty years ago, in a seminar paper [11] Nesterov proposed an accelerated gradient method, which may take the following form: starting with $x_0$ and $y_0 = x_0$, inductively define

$$x_k = y_{k-1} - s\nabla f(y_{k-1})$$
$$y_k = x_k + \frac{k-1}{k+2}(x_k - x_{k-1}). \tag{1.1}$$

For a fixed step size $s = 1/L$, where $L$ is the Lipschitz constant of $\nabla f$, this scheme exhibits the convergence rate

$$f(x_k) - f^\star \le O\Big(\frac{L\|x_0 - x^\star\|^2}{k^2}\Big).$$

Above, $x^\star$ is any minimizer of $f$ and $f^\star = f(x^\star)$. It is well-known that this rate is optimal among all methods having only information about the gradient of $f$ at consecutive iterates [12]. This is in contrast to vanilla gradient descent methods, which can only achieve a rate of $O(1/k)$ [17]. This improvement relies on the introduction of the momentum term $x_k - x_{k-1}$ as well as the particularly tuned coefficient $(k-1)/(k+2) \approx 1 - 3/k$. Since the introduction of Nesterov's scheme, there has been much work on the development of first-order accelerated methods, see [12, 13, 14, 1, 2] for example, and [19] for a unified analysis of these ideas.

In a different direction, there is a long history relating ordinary differential equations (ODE) to optimization, see [6, 4, 8, 18] for references. The connection between ODEs and numerical optimization is often established via taking step sizes to be very small so that the trajectory or solution path converges to a curve modeled by an ODE. The conciseness and well-established theory of ODEs provide deeper insights into optimization, which has led to many interesting findings [5, 7, 16].

In this work, we derive a second-order ordinary differential equation, which is the exact limit of Nesterov's scheme by taking small step sizes in (1.1). This ODE reads

$$\ddot{X} + \frac{3}{t}\dot{X} + \nabla f(X) = 0 \tag{1.2}$$

for $t > 0$, with initial conditions $X(0) = x_0$, $\dot{X}(0) = 0$; here, $x_0$ is the starting point in Nesterov's scheme, $\dot{X}$ denotes the time derivative or velocity $\mathrm{d}X/\mathrm{d}t$ and similarly $\ddot{X} = \mathrm{d}^2 X/\mathrm{d}t^2$ denotes the acceleration. The time parameter in this ODE is related to the step size in (1.1) via $t \approx k\sqrt{s}$. Case studies are provided to demonstrate that the homogeneous and conceptually simpler ODE can serve as a tool for analyzing and generalizing Nesterov's scheme. To the best of our knowledge, this work is the first to model Nesterov's scheme or its variants by ODEs.

We denote by $\mathcal{F}_L$ the class of convex functions $f$ with $L$–Lipschitz continuous gradients defined on $\mathbb{R}^n$, i.e., $f$ is convex, continuously differentiable, and obeys

$$\|\nabla f(x) - \nabla f(y)\| \leq L\|x - y\|$$

for any $x, y \in \mathbb{R}^n$, where $\|\cdot\|$ is the standard Euclidean norm and $L > 0$ is the Lipschitz constant throughout this paper. Next, $\mathcal{S}_\mu$ denotes the class of $\mu$–strongly convex functions $f$ on $\mathbb{R}^n$ with continuous gradients, i.e., $f$ is continuously differentiable and $f(x) - \mu\|x\|^2/2$ is convex. Last, we set $\mathcal{S}_{\mu,L} = \mathcal{F}_L \cap \mathcal{S}_\mu$.

## 2 Derivation of the ODE

Assume $f \in \mathcal{F}_L$ for $L > 0$. Combining the two equations of (1.1) and applying a rescaling give

$$\frac{x_{k+1} - x_k}{\sqrt{s}} = \frac{k-1}{k+2}\frac{x_k - x_{k-1}}{\sqrt{s}} - \sqrt{s}\nabla f(y_k). \tag{2.1}$$

Introduce the *ansatz* $x_k \approx X(k\sqrt{s})$ for some smooth curve $X(t)$ defined for $t \geq 0$. For fixed $t$, as the step size $s$ goes to zero, $X(t) \approx x_{t/\sqrt{s}} = x_k$ and $X(t + \sqrt{s}) \approx x_{(t+\sqrt{s})/\sqrt{s}} = x_{k+1}$ with $k = t/\sqrt{s}$. With these approximations, we get Taylor expansions:

$$(x_{k+1} - x_k)/\sqrt{s} = \dot{X}(t) + \frac{1}{2}\ddot{X}(t)\sqrt{s} + o(\sqrt{s})$$

$$(x_k - x_{k-1})/\sqrt{s} = \dot{X}(t) - \frac{1}{2}\ddot{X}(t)\sqrt{s} + o(\sqrt{s})$$

$$\sqrt{s}\nabla f(y_k) = \sqrt{s}\nabla f(X(t)) + o(\sqrt{s}),$$

where in the last equality we use $y_k - X(t) = o(1)$. Thus (2.1) can be written as

$$\dot{X}(t) + \frac{1}{2}\ddot{X}(t)\sqrt{s} + o(\sqrt{s})$$

$$= \left(1 - \frac{3\sqrt{s}}{t}\right)\left(\dot{X}(t) - \frac{1}{2}\ddot{X}(t)\sqrt{s} + o(\sqrt{s})\right) - \sqrt{s}\nabla f(X(t)) + o(\sqrt{s}). \tag{2.2}$$

By comparing the coefficients of $\sqrt{s}$ in (2.2), we obtain

$$\ddot{X} + \frac{3}{t}\dot{X} + \nabla f(X) = 0$$

for $t > 0$. The first initial condition is $X(0) = x_0$. Taking $k = 1$ in (2.1) yields $(x_2 - x_1)/\sqrt{s} = -\sqrt{s}\nabla f(y_1) = o(1)$. Hence, the second initial condition is simply $\dot{X}(0) = 0$ (vanishing initial velocity). In the formulation of [1] (see also [20]), the momentum coefficient $(k - 1)/(k + 2)$ is replaced by $\theta_k(\theta_{k-1}^{-1} - 1)$, where $\theta_k$ are iteratively defined as

$$\theta_{k+1} = \frac{\sqrt{\theta_k^4 + 4\theta_k^2} - \theta_k^2}{2} \tag{2.3}$$

starting from $\theta_0 = 1$. A bit of analysis reveals that $\theta_k(\theta_{k-1}^{-1} - 1)$ asymptotically equals $1 - 3/k + O(1/k^2)$, thus leading to the same ODE as (1.1).

Classical results in ODE theory do not directly imply the existence or uniqueness of the solution to this ODE because the coefficient $3/t$ is singular at $t = 0$. In addition, $\nabla f$ is typically not analytic at $x_0$, which leads to the inapplicability of the power series method for studying singular ODEs. Nevertheless, the ODE is well posed: the strategy we employ for showing this constructs a series of ODEs approximating (1.2) and then chooses a convergent subsequence by some compactness arguments such as the Arzelá-Ascoli theorem. A proof of this theorem can be found in the supplementary material for this paper.

**Theorem 2.1.** *For any $f \in \mathcal{F}_\infty \triangleq \cup_{L>0} \mathcal{F}_L$ and any $x_0 \in \mathbb{R}^n$, the ODE (1.2) with initial conditions $X(0) = x_0, \dot{X}(0) = 0$ has a unique global solution $X \in C^2((0, \infty); \mathbb{R}^n) \cap C^1([0, \infty); \mathbb{R}^n)$.*

## 3  Equivalence between the ODE and Nesterov's scheme

We study the stable step size allowed for numerically solving the ODE in the presence of accumulated errors. The finite difference approximation of (1.2) by the forward Euler method is

$$\frac{X(t + \Delta t) - 2X(t) + X(t - \Delta t)}{\Delta t^2} + \frac{3}{t} \frac{X(t) - X(t - \Delta t)}{\Delta t} + \nabla f(X(t)) = 0, \qquad (3.1)$$

which is equivalent to

$$X(t + \Delta t) = \left(2 - \frac{3\Delta t}{t}\right) X(t) - \Delta t^2 \nabla f(X(t)) - \left(1 - \frac{3\Delta t}{t}\right) X(t - \Delta t).$$

Assuming that $f$ is sufficiently smooth, for small perturbations $\delta x$, $\nabla f(x + \delta x) \approx \nabla f(x) + \nabla^2 f(x)\delta x$, where $\nabla^2 f(x)$ is the Hessian of $f$ evaluated at $x$. Identifying $k = t/\Delta t$, the characteristic equation of this finite difference scheme is approximately

$$\det\left(\lambda^2 - \left(2 - \Delta t^2 \nabla^2 f - \frac{3\Delta t}{t}\right)\lambda + 1 - \frac{3\Delta t}{t}\right) = 0. \qquad (3.2)$$

The numerical stability of (3.1) with respect to accumulated errors is equivalent to this: all the roots of (3.2) lie in the unit circle [9]. When $\nabla^2 f \preceq L I_n$ (i.e., $L I_n - \nabla^2 f$ is positive semidefinite), if $\Delta t/t$ small and $\Delta t < 2/\sqrt{L}$, we see that all the roots of (3.2) lie in the unit circle. On the other hand, if $\Delta t > 2/\sqrt{L}$, (3.2) can possibly have a root $\lambda$ outside the unit circle, causing numerical instability. Under our identification $s = \Delta t^2$, a step size of $s = 1/L$ in Nesterov's scheme (1.1) is approximately equivalent to a step size of $\Delta t = 1/\sqrt{L}$ in the forward Euler method, which is stable for numerically integrating (3.1).

As a comparison, note that the corresponding ODE for gradient descent with updates $x_{k+1} = x_k - s\nabla f(x_k)$, is

$$\dot{X}(t) + \nabla f(X(t)) = 0,$$

whose finite difference scheme has the characteristic equation $\det(\lambda - (1 - \Delta t \nabla^2 f)) = 0$. Thus, to guarantee $-I_n \preceq 1 - \Delta t \nabla^2 f \preceq I_n$ in worst case analysis, one can only choose $\Delta t \leq 2/L$ for a fixed step size, which is much smaller than the step size $2/\sqrt{L}$ for (3.1) when $\nabla f$ is very variable, i.e., $L$ is large.

Next, we exhibit approximate equivalence between the ODE and Nesterov's scheme in terms of convergence rates. We first recall the original result from [11].

**Theorem 3.1** (Nesterov)**.** *For any $f \in \mathcal{F}_L$, the sequence $\{x_k\}$ in (1.1) with step size $s \leq 1/L$ obeys*

$$f(x_k) - f^\star \leq \frac{2\|x_0 - x^\star\|^2}{s(k + 1)^2}.$$

Our first result indicates that the trajectory of ODE (1.2) closely resembles the sequence $\{x_k\}$ in terms of the convergence rate to a minimizer $x^\star$.

**Theorem 3.2.** *For any $f \in \mathcal{F}_\infty$, let $X(t)$ be the unique global solution to (1.2) with initial conditions $X(0) = x_0, \dot{X}(0) = 0$. For any $t > 0$,*

$$f(X(t)) - f^\star \leq \frac{2\|x_0 - x^\star\|^2}{t^2}.$$

*Proof of Theorem 3.2.* Consider the energy functional defined as

$$\mathcal{E}(t) \triangleq t^2(f(X(t)) - f^\star) + 2\|X + \frac{t}{2}\dot{X} - x^\star\|^2,$$

whose time derivative is

$$\dot{\mathcal{E}} = 2t(f(X) - f^\star) + t^2\langle\nabla f, \dot{X}\rangle + 4\langle X + \frac{t}{2}\dot{X} - x^\star, \frac{3}{2}\dot{X} + \frac{t}{2}\ddot{X}\rangle. \tag{3.3}$$

Substituting $3\dot{X}/2 + t\ddot{X}/2$ with $-t\nabla f(X)/2$, (3.3) gives

$$\dot{\mathcal{E}} = 2t(f(X) - f^\star) + 4\langle X - x^\star, -\frac{t}{2}\nabla f(X)\rangle = 2t(f(X) - f^\star) - 2t\langle X - x^\star, \nabla f(X)\rangle \le 0,$$

where the inequality follows from the convexity of $f$. Hence by monotonicity of $\mathcal{E}$ and non-negativity of $2\|X + t\dot{X}/2 - x^\star\|^2$, the gap obeys $f(X(t)) - f^\star \le \mathcal{E}(t)/t^2 \le \mathcal{E}(0)/t^2 = 2\|x_0 - x^\star\|^2/t^2$.

$\square$

# 4 A family of generalized Nesterov's schemes

In this section we show how to exploit the power of the ODE for deriving variants of Nesterov's scheme. One would be interested in studying the ODE (1.2) with the number 3 appearing in the coefficient of $\dot{X}/t$ replaced by a general constant $r$ as in

$$\ddot{X} + \frac{r}{t}\dot{X} + \nabla f(X) = 0, \ X(0) = x_0, \dot{X}(0) = 0. \tag{4.1}$$

Using arguments similar to those in the proof of Theorem 2.1, this new ODE is guaranteed to assume a unique global solution for any $f \in \mathcal{F}_\infty$.

## 4.1 Continuous optimization

To begin with, we consider a modified energy functional defined as

$$\mathcal{E}(t) = \frac{2t^2}{r-1}(f(X(t)) - f^\star) + (r-1)\left\|X(t) + \frac{t}{r-1}\dot{X}(t) - x^\star\right\|^2.$$

Since $r\dot{X} + t\ddot{X} = -t\nabla f(X)$, the time derivative $\dot{\mathcal{E}}$ is equal to

$$\frac{4t}{r-1}(f(X) - f^\star) + \frac{2t^2}{r-1}\langle\nabla f, \dot{X}\rangle + 2\langle X + \frac{t}{r-1}\dot{X} - x^\star, r\dot{X} + t\ddot{X}\rangle$$
$$= \frac{4t}{r-1}(f(X) - f^\star) - 2t\langle X - x^\star, \nabla f(X)\rangle. \tag{4.2}$$

A consequence of (4.2) is this:

**Theorem 4.1.** *Suppose $r > 3$ and let $X$ be the unique solution to (4.1) for some $f \in \mathcal{F}_\infty$. Then $X$ obeys*

$$f(X(t)) - f^\star \le \frac{(r-1)^2\|x_0 - x^\star\|^2}{2t^2}$$

*and*

$$\int_0^\infty t(f(X(t)) - f^\star)\mathrm{d}t \le \frac{(r-1)^2\|x_0 - x^\star\|^2}{2(r-3)}.$$

*Proof of Theorem 4.1.* By (4.2), the derivative $\mathrm{d}\mathcal{E}/\mathrm{d}t$ equals

$$2t(f(X) - f^\star) - 2t\langle X - x^\star, \nabla f(X)\rangle - \frac{2(r-3)t}{r-1}(f(X) - f^\star) \le -\frac{2(r-3)t}{r-1}(f(X) - f^\star), \tag{4.3}$$

where the inequality follows from the convexity of $f$. Since $f(X) \ge f^\star$, (4.3) implies that $\mathcal{E}$ is non-increasing. Hence

$$\frac{2t^2}{r-1}(f(X(t)) - f^\star) \le \mathcal{E}(t) \le \mathcal{E}(0) = (r-1)\|x_0 - x^\star\|^2,$$

yielding the first inequality of the theorem as desired. To complete the proof, by (4.2) it follows that

$$\int_0^\infty \frac{2(r-3)t}{r-1}(f(X) - f^\star)\mathrm{d}t \leq -\int_0^\infty \frac{\mathrm{d}\mathcal{E}}{\mathrm{d}t}\mathrm{d}t = \mathcal{E}(0) - \mathcal{E}(\infty) \leq (r-1)\|x_0 - x^\star\|^2,$$

as desired for establishing the second inequality. □

We now demonstrate faster convergence rates under the assumption of strong convexity. Given a strongly convex function $f$, consider a new energy functional defined as

$$\tilde{\mathcal{E}}(t) = t^3(f(X(t)) - f^\star) + \frac{(2r-3)^2 t}{8}\left\|X(t) + \frac{2t}{2r-3}\dot{X}(t) - x^\star\right\|^2.$$

As in Theorem 4.1, a more refined study of the derivative of $\tilde{\mathcal{E}}(t)$ gives

**Theorem 4.2.** *For any $f \in \mathcal{S}_{\mu,L}(\mathbb{R}^n)$, the unique solution $X$ to (4.1) with $r \geq 9/2$ obeys*

$$f(X(t)) - f^\star \leq \frac{Cr^{\frac{5}{2}}\|x_0 - x^\star\|^2}{t^3\sqrt{\mu}}$$

*for any $t > 0$ and a universal constant $C > 1/2$.*

The restriction $r \geq 9/2$ is an artifact required in the proof. We believe that this theorem should be valid as long as $r \geq 3$. For example, the solution to (4.1) with $f(x) = \|x\|^2/2$ is

$$X(t) = \frac{2^{\frac{r-1}{2}}\Gamma((r+1)/2)J_{(r-1)/2}(t)}{t^{\frac{r-1}{2}}}x_0, \tag{4.4}$$

where $J_{(r-1)/2}(\cdot)$ is the first kind Bessel function of order $(r-1)/2$. For large $t$, this Bessel function obeys $J_{(r-1)/2}(t) = \sqrt{2/(\pi t)}(\cos(t - (r-1)\pi/4 - \pi/4) + O(1/t))$. Hence,

$$f(X(t)) - f^\star \lesssim \|x_0 - x^\star\|^2/t^r,$$

in which the inequality fails if $1/t^r$ is replaced by any higher order rate. For general strongly convex functions, such refinement, if possible, might require a construction of a more sophisticated energy functional and careful analysis. We leave this problem for future research.

## 4.2 Composite optimization

Inspired by Theorem 4.2, it is tempting to obtain such analogies for the discrete Nesterov's scheme as well. Following the formulation of [1], we consider the composite minimization:

$$\underset{x \in \mathbb{R}^n}{\text{minimize}} \quad f(x) = g(x) + h(x),$$

where $g \in \mathcal{F}_L$ for some $L > 0$ and $h$ is convex on $\mathbb{R}^n$ with possible extended value $\infty$. Define the proximal subgradient

$$G_s(x) \triangleq \frac{x - \text{argmin}_z\left[\|z - (x - s\nabla g(x))\|^2/(2s) + h(z)\right]}{s}.$$

Parametrizing by a constant $r$, we propose a generalized Nesterov's scheme,

$$\begin{aligned} x_k &= y_{k-1} - sG_s(y_{k-1}) \\ y_k &= x_k + \frac{k-1}{k+r-1}(x_k - x_{k-1}), \end{aligned} \tag{4.5}$$

starting from $y_0 = x_0$. The discrete analog of Theorem 4.1 is below, whose proof is also deferred to the supplementary materials as well.

**Theorem 4.3.** *The sequence $\{x_k\}$ given by (4.5) with $r > 3$ and $0 < s \leq 1/L$ obeys*

$$f(x_k) - f^\star \leq \frac{(r-1)^2\|x_0 - x^\star\|^2}{2s(k+r-2)^2}$$

*and*

$$\sum_{k=1}^\infty (k+r-1)(f(x_k) - f^\star) \leq \frac{(r-1)^2\|x_0 - x^\star\|^2}{2s(r-3)}.$$

The idea behind the proof is the same as that employed for Theorem 4.1; here, however, the energy functional is defined as

$$\mathcal{E}(k) = 2s(k + r - 2)^2(f(x_k) - f^\star)/(r - 1) + \|(k + r - 1)y_k - kx_k - (r - 1)x^\star\|^2/(r - 1).$$

The first inequality in Theorem 4.3 suggests that the generalized Nesterov's scheme still achieves $O(1/k^2)$ convergence rate. However, if the error bound satisfies

$$f(x_{k'}) - f^\star \geq \frac{c}{k'^2}$$

for some $c > 0$ and a dense subsequence $\{k'\}$, i.e., $|\{k'\} \cap \{1, \ldots, m\}| \geq \alpha m$ for any positive integer $m$ and some $\alpha > 0$, then the second inequality of the theorem is violated. Hence, the second inequality is not trivial because it implies the error bound is in some sense $O(1/k^2)$ suboptimal.

In closing, we would like to point out this new scheme is equivalent to setting $\theta_k = (r-1)/(k+r-1)$ and letting $\theta_k(\theta_{k-1}^{-1} - 1)$ replace the momentum coefficient $(k - 1)/(k + r - 1)$. Then, the equal sign " $=$ " in (2.3) has to be replaced by " $\geq$ ". In examining the proof of Theorem 1(b) in [20], we can get an alternative proof of Theorem 4.3 by allowing (2.3), which appears in Eq. (36) in [20], to be an inequality.

## 5  Accelerating to linear convergence by restarting

Although an $O(1/k^3)$ convergence rate is guaranteed for generalized Nesterov's schemes (4.5), the example (4.4) provides evidence that $O(1/\texttt{poly}(k))$ is the best rate achievable under strong convexity. In contrast, the vanilla gradient method achieves linear convergence $O((1 - \mu/L)^k)$ and [12] proposed a first-order method with a convergence rate of $O((1 - \sqrt{\mu/L})^k)$, which, however, requires knowledge of the condition number $\mu/L$. While it is relatively easy to bound the Lipschitz constant $L$ by the use of backtracking [3, 19], estimating the strong convexity parameter $\mu$, if not impossible, is very challenging. Among many approaches to gain acceleration via adaptively estimating $\mu/L$, [15] proposes a restarting procedure for Nesterov's scheme in which (1.1) is restarted with $x_0 = y_0 := x_k$ whenever $\nabla f(y_k)^T(x_{k+1} - x_k) > 0$. In the language of ODEs, this gradient based restarting essentially keeps $\langle \nabla f, \dot{X} \rangle$ negative along the trajectory. Although it has been empirically observed that this method significantly boosts convergence, there is no general theory characterizing the convergence rate.

In this section, we propose a new restarting scheme we call the *speed restarting scheme*. The underlying motivation is to maintain a relatively high velocity $\dot{X}$ along the trajectory. Throughout this section we assume $f \in \mathcal{S}_{\mu,L}$ for some $0 < \mu \leq L$.

**Definition 5.1.** *For ODE* (1.2) *with* $X(0) = x_0$, $\dot{X}(0) = 0$, *let*

$$T = T(f, x_0) = \sup\{t > 0 : \forall u \in (0, t), \frac{\mathrm{d}\|\dot{X}(u)\|^2}{\mathrm{d}u} > 0\}$$

*be the speed restarting time.*

In words, $T$ is the first time the velocity $\|\dot{X}\|$ decreases. The definition itself does not imply that $0 < T < \infty$, which is proven in the supplementary materials. Indeed, $f(X(t))$ is a decreasing function before time $T$; for $t \leq T$,

$$\frac{\mathrm{d}f(X(t))}{\mathrm{d}t} = \langle \nabla f(X), \dot{X} \rangle = -\frac{3}{t}\|\dot{X}\|^2 - \frac{1}{2}\frac{\mathrm{d}\|\dot{X}\|^2}{\mathrm{d}t} \leq 0.$$

The speed restarted ODE is thus

$$\ddot{X}(t) + \frac{3}{t_{\mathrm{sr}}}\dot{X}(t) + \nabla f(X(t)) = 0, \tag{5.1}$$

where $t_{\mathrm{sr}}$ is set to zero whenever $\langle \dot{X}, \ddot{X} \rangle = 0$ and between two consecutive restarts, $t_{\mathrm{sr}}$ grows just as $t$. That is, $t_{\mathrm{sr}} = t - \tau$, where $\tau$ is the latest restart time. In particular, $t_{\mathrm{sr}} = 0$ at $t = 0$. The theorem below guarantees linear convergence of the solution to (5.1). This is a new result in the literature [15, 10].

**Theorem 5.2.** *There exists positive constants $c_1$ and $c_2$, which only depend on the condition number $L/\mu$, such that for any $f \in \mathcal{S}_{\mu,L}$, we have*

$$f(X^{\mathrm{sr}}(t)) - f(x^\star) \leq \frac{c_1 L\|x_0 - x^\star\|^2}{2}\mathrm{e}^{-c_2 t\sqrt{L}}.$$

## 5.1 Numerical examples

Below we present a discrete analog to the restarted scheme. There, $k_{\min}$ is introduced to avoid having consecutive restarts that are too close. To compare the performance of the restarted scheme with the original (1.1), we conduct four simulation studies, including both smooth and non-smooth objective functions. Note that the computational costs of the restarted and non-restarted schemes are the same.

---

**Algorithm 1** Speed Restarting Nesterov's Scheme

---

**input:** $x_0 \in \mathbb{R}^n, y_0 = x_0, x_{-1} = x_0, 0 < s \leq 1/L, k_{\max} \in \mathbb{N}^+$ and $k_{\min} \in \mathbb{N}^+$
$j \leftarrow 1$
**for** $k = 1$ to $k_{\max}$ **do**
    $x_k \leftarrow \text{argmin}_x(\frac{1}{2s}\|x - y_{k-1} + s\nabla g(y_{k-1})\|^2 + h(x))$
    $y_k \leftarrow x_k + \frac{j-1}{j+2}(x_k - x_{k-1})$
    **if** $\|x_k - x_{k-1}\| < \|x_{k-1} - x_{k-2}\|$ **and** $j \geq k_{\min}$ **then**
        $j \leftarrow 1$
    **else**
        $j \leftarrow j + 1$
    **end if**
**end for**

---

**Quadratic.** $f(x) = \frac{1}{2}x^T A x + b^T x$ is a strongly convex function, in which $A$ is a $500 \times 500$ random positive definite matrix and $b$ a random vector. The eigenvalues of $A$ are between 0.001 and 1. The vector $b$ is generated as i. i. d. Gaussian random variables with mean 0 and variance 25.

**Log-sum-exp.**

$$f(x) = \rho \log \Big[ \sum_{i=1}^m \exp((a_i^T x - b_i)/\rho) \Big],$$

where $n = 50, m = 200, \rho = 20$. The matrix $A = \{a_{ij}\}$ is a random matrix with i. i. d. standard Gaussian entries, and $b = \{b_i\}$ has i. i. d. Gaussian entries with mean 0 and variance 2. This function is not strongly convex.

**Matrix completion.** $f(X) = \frac{1}{2}\|X_{\text{obs}} - M_{\text{obs}}\|_F^2 + \lambda\|X\|_*$, in which the ground truth $M$ is a rank-5 random matrix of size $300 \times 300$. The regularization parameter is set to $\lambda = 0.05$. The 5 singular values of $M$ are $1, \ldots, 5$. The observed set is independently sampled among the $300 \times 300$ entries so that 10% of the entries are actually observed.

**Lasso in $\ell_1$–constrained form with large sparse design.** $f = \frac{1}{2}\|Ax - b\|^2$   s.t. $\|x\|_1 \leq \delta$, where $A$ is a $5000 \times 50000$ random sparse matrix with nonzero probability 0.5% for each entry and $b$ is generated as $b = Ax^0 + z$. The nonzero entries of $A$ independently follow the Gaussian distribution with mean 0 and variance $1/25$. The signal $x^0$ is a vector with 250 nonzeros and $z$ is i. i. d. standard Gaussian noise. The parameter $\delta$ is set to $\|x^0\|_1$.

In these examples, $k_{\min}$ is set to be 10 and the step sizes are fixed to be $1/L$. If the objective is in composite form, the Lipschitz bound applies to the smooth part. Figures 1(a), 1(b), 1(c) and 1(d) present the performance of the speed restarting scheme, the gradient restarting scheme proposed in [15], the original Nesterov's scheme and the proximal gradient method. The objective functions include strongly convex, non-strongly convex and non-smooth functions, violating the assumptions in Theorem 5.2. Among all the examples, it is interesting to note that both speed restarting scheme empirically exhibit linear convergence by significantly reducing bumps in the objective values. This leaves us an open problem of whether there exists provable linear convergence rate for the gradient restarting scheme as in Theorem 5.2. It is also worth pointing that compared with gradient restarting, the speed restarting scheme empirically exhibits more stable linear convergence rate.

## 6 Discussion

This paper introduces a second-order ODE and accompanying tools for characterizing Nesterov's accelerated gradient method. This ODE is applied to study variants of Nesterov's scheme. Our

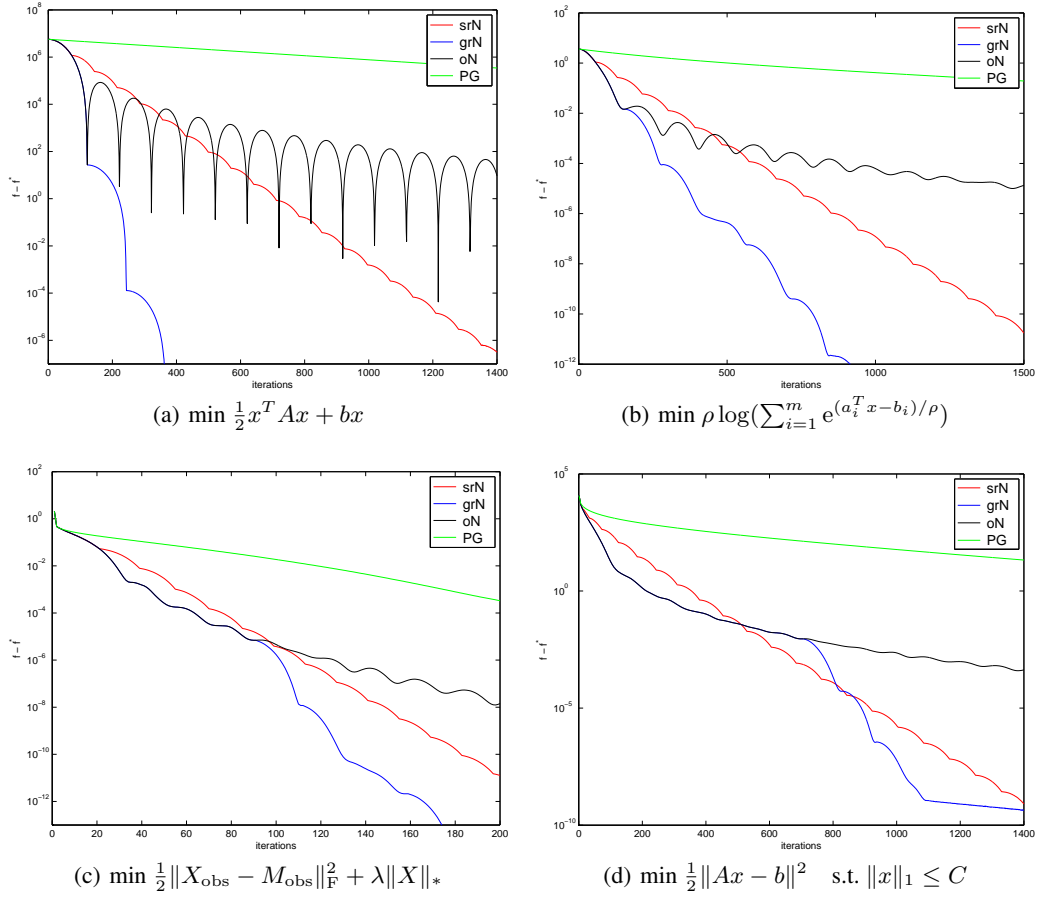

Figure 1: Numerical performance of speed restarting (srN), gradient restarting (grN) proposed in [15], the original Nesterov's scheme (oN) and the proximal gradient (PG)

approach suggests (1) a large family of generalized Nesterov's schemes that are all guaranteed to converge at the rate $1/k^2$, and (2) a restarted scheme provably achieving a linear convergence rate whenever $f$ is strongly convex.

In this paper, we often utilize ideas from continuous-time ODEs, and then apply these ideas to discrete schemes. The translation, however, involves parameter tuning and tedious calculations. This is the reason why a general theory mapping properties of ODEs into corresponding properties for discrete updates would be a welcome advance. Indeed, this would allow researchers to only study the simpler and more user-friendly ODEs.

# 7 Acknowledgements

We would like to thank Carlos Sing-Long and Zhou Fan for helpful discussions about parts of this paper, and anonymous reviewers for their insightful comments and suggestions.

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
