[Supplementary Material]

# Supplementary materials for *A Differential Equation for Modeling Nesterov's Accelerated Gradient Method: Theory and Insights*

**Weijie Su**
Department of Statistics
Stanford University
Stanford, CA 94305
wjsu@stanford.edu

**Stephen Boyd**
Department of Electrical Engineering
Stanford University
Stanford, CA 94305
boyd@stanford.edu

**Emmanuel J. Candès**
Departments of Statistics and of Mathematics
Stanford University
Stanford, CA 94305
candes@stanford.edu

## 1 Proof of Theorem 2.1

The proof is divided into two parts, namely, existence and uniqueness.

### 1.1 Existence

In this section we aim to prove

**Lemma 1.1.** *For any $f \in \mathcal{F}_\infty(\mathbb{R}^n)$ and any $x_0 \in \mathbb{R}^n$, ODE (1.2) with initial conditions $X(0) = x_0, \dot{X}(0) = 0$ has at least one solution $X$ in $C^2(0,\infty) \cap C^1[0,\infty)$. Recall $C^2(0,\infty)$ is the set of functions, taking values in $\mathbb{R}^n$, defined on $[0,\infty)$ and twice continuously differentiable on $(0,\infty)$. Similarly $C^1[0,\infty)$ is the set of continuously differentiable functions from $[0,\infty)$ to $\mathbb{R}^n$.*

To begin with, for any $\delta > 0$ consider the smoothed ODE

$$\ddot{X} + \frac{3}{\max(\delta,t)}\dot{X} + \nabla f(X) = 0 \tag{1}$$

with $X(0) = x_0, \dot{X}(0) = 0$. Denoting by $Z = \dot{X}$, then (1) is equivalent to

$$\frac{d}{dt}\begin{pmatrix} X \\ Z \end{pmatrix} = \begin{pmatrix} Z \\ -\frac{3}{\max(\delta,t)}Z - \nabla f(X) \end{pmatrix}.$$

As functions of $(X, Z)$, both ($Z$ and $-3Z/\max(\delta,t) - \nabla f(X)$) are Lipschitz continuous with constant at most $\max(1, L) + 3/\delta$. Hence by standard ODE theory (1) has a unique global solution in $C^2[0,\infty)$, which is denoted by $X_\delta$. Note that $\ddot{X}_\delta$ is also well defined at $t = 0$. Next, introduce $M_\delta(t)$ to be the supremum of $\|\dot{X}_\delta(u)\|/u$ over $u \in (0, t]$. We remark that $M_\delta(t)$ is finite because $\|\dot{X}_\delta(u)\|/u = (\|\dot{X}_\delta(u) - \dot{X}_\delta(0)\|)/u = \|\ddot{X}_\delta(0)\| + o(1)$ for $u = o(1)$. We given an upper bound for $M_\delta(t)$ in the following lemma.

**Lemma 1.2.** *For $\delta < \sqrt{6/L}$ one has*

$$M_\delta(\delta) \le \frac{\|\nabla f(x_0)\|}{1 - L\delta^2/6}.$$

The proof of Lemma 1.2 relies on a simple lemma.

**Lemma 1.3.** *For any $u > 0$, the following inequality holds*

$$\|\nabla f(X_\delta(u)) - \nabla f(x_0)\| \leq \frac{1}{2} L M_\delta(u) u^2.$$

*Proof of Lemma 1.3.* By Lipschitz continuity,

$$\|\nabla f(X_\delta(u)) - \nabla f(x_0)\| \leq L\|X_\delta(u) - x_0\| = \left\| \int_0^u \dot{X}_\delta(v) dv \right\| \leq \int_0^u v \frac{\|\dot{X}_\delta(v)\|}{v} dv \leq \frac{1}{2} L M_\delta(u) u^2.$$

$\square$

*Proof of Lemma 1.2.* For $0 < t \leq \delta$, the smoothed ODE reads

$$\ddot{X}_\delta + \frac{3}{\delta} \dot{X}_\delta + \nabla f(X_\delta) = 0,$$

which yields

$$\dot{X}_\delta e^{3t/\delta} = -\int_0^t \nabla f(X_\delta(u)) e^{3u/\delta} du = -\nabla f(x_0) \int_0^t e^{3u/\delta} du - \int_0^t (\nabla f(X_\delta(u)) - \nabla f(x_0)) e^{3u/\delta} du.$$

Hence, by Lemma 1.3

$$\frac{\|\dot{X}_\delta(t)\|}{t} \leq \frac{1}{t} e^{-3t/\delta} \|\nabla f(x_0)\| \int_0^t e^{3u/\delta} du + \frac{1}{t} e^{-3t/\delta} \int_0^t \frac{1}{2} L M_\delta(u) u^2 e^{3u/\delta} du$$

$$\leq \|\nabla f(x_0)\| + \frac{L M_\delta(\delta) \delta^2}{6}.$$

Taking the supremum of $\|\dot{X}_\delta(t)\|/t$ over $0 < t \leq \delta$ and rearranging the inequality give the desired result. $\square$

Next, we give an upper bound for $M_\delta(t)$ with $t > \delta$.

**Lemma 1.4.** *For $\delta < \sqrt{6/L}$ and $\delta < t < \sqrt{12/L}$, one has*

$$M_\delta(t) \leq \frac{(5 - L\delta^2/6)\|\nabla f(x_0)\|}{4(1 - L\delta^2/6)(1 - Lt^2/12)}.$$

*Proof of Lemma 1.4.* When $t > \delta$ the smoothed ODE reads

$$\ddot{X}_\delta + \frac{3}{t} \dot{X}_\delta + \nabla f(X_\delta) = 0,$$

which is equivalent to

$$\frac{d t^3 \dot{X}_\delta(t)}{dt} = -t^3 \nabla f(X_\delta(t)).$$

By integration,

$$t^3 \dot{X}_\delta(t) = -\int_\delta^t u^3 \nabla f(X_\delta(u)) du + \delta^3 \dot{X}_\delta(\delta) = -\int_\delta^t u^3 \nabla f(x_0) du - \int_\delta^t u^3 (\nabla f(X_\delta(u)) - \nabla f(x_0)) du + \delta^3 \dot{X}_\delta(\delta).$$

Therefore by Lemmas 1.3 and 1.2 we have

$$\frac{\|\dot{X}_\delta(t)\|}{t} \leq \frac{t^4 - \delta^4}{4t^4} \|\nabla f(x_0)\| + \frac{1}{t^4} \int_\delta^t \frac{1}{2} L M_\delta(u) u^5 du + \frac{\delta^4}{t^4} \frac{\|\dot{X}_\delta(\delta)\|}{\delta}$$

$$\leq \frac{1}{4} \|\nabla f(x_0)\| + \frac{1}{12} L M_\delta(t) t^2 + \frac{\|\nabla f(X_0)\|}{1 - L\delta^2/6},$$

where the last expression is an increasing function of $t$. So for any $\delta < t' < t$, it follows that

$$\frac{\|\dot{X}_\delta(t')\|}{t'} \leq \frac{1}{4} \|\nabla f(x_0)\| + \frac{1}{12} L M_\delta(t) t^2 + \frac{\|\nabla f(x_0)\|}{1 - L\delta^2/6},$$

which also holds for $t' \leq \delta$. Taking the supremum over $t' \in (0, t)$ gives

$$M_\delta(t) \leq \frac{1}{4}\|\nabla f(x_0)\| + \frac{1}{12}LM_\delta(t)t^2 + \frac{\|\nabla f(X_0)\|}{1 - L\delta^2/6}.$$

The desired result follows from rearranging the inequality. $\qquad \square$

**Lemma 1.5.** *Consider the set of continuous functions* $\mathcal{F} = \{X_\delta : [0, \sqrt{6/L}] \to \mathbb{R}^n | \delta = \sqrt{3/L}/2^m, m = 0, 1, \ldots\}$ *is uniformly bounded and equicontinuous.*

*Proof of Lemma 1.5.* By Lemmas 1.2 and 1.4, for any $t \in [0, \sqrt{6/L}], \delta \in (0, \sqrt{3/L})$ the gradient is uniformly bounded by

$$\|\dot{X}_\delta(t)\| \leq \sqrt{6/L}M_\delta(\sqrt{6/L}) \leq \sqrt{6/L}\max\left\{\frac{\|\nabla f(x_0)\|}{1 - \frac{1}{2}}, \frac{5\|\nabla f(x_0)\|}{4(1 - \frac{1}{2})(1 - \frac{1}{2})}\right\} = 5\sqrt{6/L}\|\nabla f(x_0)\|.$$

Thus it immediately implies that $\mathcal{F}$ is equicontinuous. To establish the uniform boundedness, note that

$$\|X_\delta(t)\| \leq \|X_\delta(0)\| + \int_0^t \|\dot{X}_\delta(u)\|du \leq \|x_0\| + 30\|\nabla f(x_0)\|/L.$$

$\qquad \square$

Now it is ready to give

*Proof of Lemma 1.1.* By the Arzelá–Ascoli theorem and Lemma 1.5, $\mathcal{F}$ contains a sequence converge uniformly on $[0, \sqrt{6/L}]$. Denote by $\{X_{\delta_{m_i}}\}_{i\in\mathbb{N}}$ the convergent sequence and $\check{X}$ the limit. Above, $\delta_{m_i} = \sqrt{3/L}/2^{m_i}$ decreases as $i$ increases. We will prove that $\check{X}$ satisfies (1.2) and the initial conditions $\check{X}(0) = x_0, \dot{\check{X}}(0) = 0$.

Fix an arbitrary $t_0 \in (0, \sqrt{6/L})$. Since $\|\dot{X}_{\delta_{m_i}}(t_0)\|$ is bounded, we can pick a subsequence of $\dot{X}_{\delta_{m_i}}(t_0)$ which converges to a limit denoted by $X_{t_0}^D$. Without loss of generality, assume the subsequence is the original sequence. Denote by $\tilde{X}$ the local solution to (1.2) with $X(t_0) = \check{X}(t_0)$ and $\dot{X}(t_0) = X_{t_0}^D$. On the other hand, recall $X_{\delta_{m_i}}$ is the solution to (1.2) with $X(t_0) = X_{\delta_{m_i}}(t_0)$ and $\dot{X}(t_0) = \dot{X}_{\delta_{m_i}}(t_0)$ when $\delta_{m_i} < t_0$. Since both $X_{\delta_{m_i}}(t_0)$ and $\dot{X}_{\delta_{m_i}}(t_0)$ go to $\check{X}(t_0)$ and $X_{t_0}^D$, respectively, there exits $\epsilon_0 > 0$ such that

$$\sup_{t\in(t_0-\epsilon_0,t_0+\epsilon_0)} \|X_{\delta_{m_i}}(t) - \tilde{X}(t)\| \to 0$$

as $i \to \infty$. However, by definition we have

$$\sup_{t\in(t_0-\epsilon_0,t_0+\epsilon_0)} \|X_{\delta_{m_i}}(t) - \check{X}(t)\| \to 0.$$

Therefore $\check{X}$ and $\tilde{X}$ have to be identical on $(t_0 - \epsilon_0, t_0 + \epsilon_0)$. So $\check{X}$ satisfies (1.2) at $t_0$. Since $t_0$ is arbitrary, we conclude that $\check{X}$ is a solution to (1.2) on $(0, \sqrt{6/L})$. By extension, $\check{X}$ can be a global solution to (1.2) on $(0, \infty)$. It only leaves to verify the initial conditions to complete the proof.

The first condition $\check{X}(0) = x_0$ is a direct consequence of $X_{\delta_{m_i}}(0) = x_0$. To check the second one, pick a small $t > 0$ and note that

$$\frac{\|\check{X}(t) - \check{X}(0)\|}{t} = \lim_{i\to\infty}\frac{\|X_{\delta_{m_i}}(t) - X_{\delta_{m_i}}(0)\|}{t} = \lim_{i\to\infty}\|\dot{X}_{\delta_{m_i}}(\xi_i)\| \leq \limsup_{i\to\infty} tM_{\delta_{m_i}}(t) \leq 5t\sqrt{6/L}\|\nabla f(x_0)\|,$$

where $\xi_i \in (0, t)$ is by the mean value theorem. The desired result follows from taking $t \to 0$. $\quad \square$

## 1.2 Uniqueness

In this section we prove the uniqueness of the solution to (1.2).

**Lemma 1.6.** *For any initial point $x_0 \in \mathbb{R}^n$, ODE (1.2) with initial conditions $X(0) = x_0, \dot{X}(0) = 0$ has at most one local solution near $t = 0$.*

Suppose on the contrary there are two solutions, namely, $X$ and $Y$ defined on $(0, \alpha)$ for some $\alpha > 0$. Define $\tilde{M}(t)$ to be the supremum of $\|\dot{X}(u) - \dot{Y}(u)\|$ over $u \in [0, t)$, where $t$ is between $\epsilon$ and $\alpha$. To proceed, we need a simple auxiliary lemma.

**Lemma 1.7.** *For any $t \in (0, \alpha)$ one has*
$$\|\nabla f(X(t)) - \nabla f(Y(t))\| \leq Lt\tilde{M}(t).$$

*Proof of Lemma 1.7.* By Lipschitz continuity of the gradient, one has

$$\|\nabla f(X(t)) - \nabla f(Y(t))\| \leq L\|X(t) - Y(t)\| = L \left\| \int_0^t \dot{X}(u) - \dot{Y}(u)du + X(0) - Y(0) \right\|$$

$$\leq L \int_0^t \|\dot{X}(u) - \dot{Y}(u)\|du \leq Lt\tilde{M}(t).$$

$\square$

*Proof of Lemma 1.6.* Similar to the proof of Lemma 1.4, one has

$$t^3(\dot{X}(t) - \dot{Y}(t)) = -\int_0^t u^3(\nabla f(X(u)) - \nabla f(Y(u)))du.$$

Applying Lemma 1.7 gives

$$t^3\|\dot{X}(t) - \dot{Y}(t)\| \leq \int_0^t Lu^4\tilde{M}(u)du \leq \frac{1}{5}Lt^5\tilde{M}(t),$$

which reads $\|\dot{X}(t) - \dot{Y}(t)\| \leq Lt^2\tilde{M}(t)/5$. Thus for any $t' \leq t$ it is true that $\|\dot{X}(t') - \dot{Y}(t')\| \leq Lt^2\tilde{M}(t)/5$. Taking the supremum of $\|\dot{X}(t') - \dot{Y}(t')\|$ over $t' \in (0, t)$ gives $\tilde{M}(t) \leq Lt^2\tilde{M}(t)/5$. Therefore $\tilde{M}(t) = 0$ for $t < \min(\alpha, \sqrt{5/L})$, which is equivalent to saying $\dot{X} = \dot{Y}$ on $[0, \min(\alpha, \sqrt{5/L}))$. With the same initial value $X(0) = Y(0) = x_0$ and the same gradient, we conclude that $X$ and $Y$ are identical on $(0, \min(\alpha, \sqrt{5/L}))$, a contradiction. $\square$

*Proof of Theorem 2.1.* Lemma 1.1 together with Lemma 1.6 completes the proof of Theorem 2.1.
$\square$

## 2   Proof of Theorem 4.2

*Proof of Theorem 4.2.* The derivative of $\tilde{\mathcal{E}}$ reads

$$\frac{d\tilde{\mathcal{E}}(t)}{dt} = 3t^2(f(X) - f^\star) + t^3\langle \dot{X}, \nabla f(X) \rangle + \frac{(2r-3)^2}{8}\left\langle X + \frac{2t}{2r-3}\dot{X} - x^\star, \frac{4t^2}{2r-3}\ddot{X} + \frac{4rt}{2r-3}\dot{X} + X - x^\star \right\rangle$$

$$= 3t^2(f(X) - f^\star) - \frac{(2r-3)t^2}{2}\langle X - x^\star, \nabla f(X) \rangle + \frac{(2r-3)^2}{8}\|X - x^\star\|^2 + \frac{(2r-3)t}{4}\langle \dot{X}, X - x^\star \rangle. \quad (2)$$

By convexity and strong convexity of $f$, the second term of the RHS of (2) meets

$$\frac{(2r-3)t^2}{2}\langle X - x^\star, \nabla f(X) \rangle \geq \frac{(2r-3)t^2}{2}(f(X) - f^\star) + \frac{\mu(2r-3)t^2}{4}\|X - x^\star\|^2.$$

Since $r \geq 4$, substituting the above into (2) yields

$$\frac{d\tilde{\mathcal{E}}(t)}{dt} \leq \left[3t^2 - \frac{(2r-3)t^2}{2}\right](f(X) - f^\star) - \frac{2(2r-3)\mu t^2 - (2r-3)^2}{8}\|X - x^\star\|^2 + \frac{(2r-3)t}{8}\frac{d\|X - x^\star\|^2}{dt}$$

$$\leq -\frac{2(2r-3)\mu t^2 - (2r-3)^2}{8}\|X - x^\star\|^2 + \frac{(2r-3)t}{8}\frac{d\|X - x^\star\|^2}{dt}.$$

Hence if $t \geq t' \triangleq \sqrt{(2r-3)/(2\mu)}$, we obtain

$$\frac{d\tilde{\mathcal{E}}(t)}{dt} \leq \frac{(2r-3)t}{8}\frac{d\|X - x^\star\|^2}{dt}. \tag{3}$$

For $t > t'$, integrating (3) over $(t', t)$ gives

$$\tilde{\mathcal{E}}(t) \leq \tilde{\mathcal{E}}(t') + \frac{2r-3}{8}t\|X(t) - x^\star\|^2 - \frac{2r-3}{8}t'\|X(t') - x^\star\|^2 - \frac{2r-3}{8}\int_{t'}^t \|X(u) - x^\star\|^2 du$$

$$\leq \tilde{\mathcal{E}}(t') + \frac{2r-3}{8}t\|X(t) - x^\star\|^2 \leq \tilde{\mathcal{E}}(t') + \frac{2r-3}{4\mu}t(f(X(t)) - f^\star)$$

$$\leq \tilde{\mathcal{E}}(t') + \frac{(2r-3)(r-1)^2\|x_0 - x^\star\|^2}{8\mu t} \leq \tilde{\mathcal{E}}(t') + \frac{(2r-3)(r-1)^2\|x_0 - x^\star\|^2}{8\mu t'}, \tag{4}$$

where the second last inequality follows from Theorem 4.1. We can make use of $\mathcal{E}(t')$ to bound $\tilde{\mathcal{E}}(t')$ in (4). Indeed we have

$$\tilde{\mathcal{E}}(t') = t'^3(f(X(t')) - f^\star) + \frac{(2r-3)^2t'}{8}\|X(t') + \frac{2t'}{2r-3}\dot{X}(t') - x^\star\|^2$$

$$\leq t'^3(f(X(t')) - f^\star) + \frac{(2r-3)^2t'}{4}\left\|\frac{2r-2}{2r-3}X(t') + \frac{2t'}{2r-3}\dot{X}(t') - \frac{2r-2}{2r-3}x^\star\right\|^2$$

$$+ \frac{(2r-3)^2t'}{4}\left\|\frac{1}{2r-3}X(t') - \frac{1}{2r-3}x^\star\right\|^2$$

$$\leq (r-1)t'\mathcal{E}(t') + \frac{t'}{4}\|X(t') - x^\star\|^2 \leq (r-1)^2t'\|x_0 - x^\star\|^2 + \frac{(r-1)^2\|x_0 - x^\star\|^2}{4\mu t'},$$

which combined with (4) yields

$$\tilde{\mathcal{E}}(t) \leq (r-1)^2t'\|x_0 - x^\star\|^2 + \frac{(2r-1)(r-1)^2\|x_0 - x^\star\|^2}{8\mu t'} = O\left(\frac{r^{\frac{5}{2}}\|x_0 - x^\star\|^2}{\sqrt{\mu}}\right).$$

It completes the proof for $t \geq \sqrt{(2r-3)/(2\mu)}$ by noting $f(X(t)) - f^\star \leq \tilde{\mathcal{E}}(t)/t^3$, whereas for $t < \sqrt{(2r-3)/(2\mu)}$ by Theorem 4.1 we have

$$f(X(t)) - f^\star \leq \frac{(r-1)^2\|x_0 - x^\star\|^2}{2t^2} \leq \frac{(r-1)^2\sqrt{\mu}\sqrt{(2r-3)/(2\mu)}}{2Cr^{\frac{5}{2}}}\frac{Cr^{\frac{5}{2}}\|x_0 - x^\star\|^2}{t^3\sqrt{\mu}} \leq \frac{Cr^{\frac{5}{2}}\|x_0 - x^\star\|^2}{t^3\sqrt{\mu}}.$$

$\square$

# 3 Proof of Theorem 4.3

*Proof of Theorem 4.3.* In parallel to the proof of Theorem 4.1, we propose an energy function defined as

$$\mathcal{E}(k) = \frac{2(k+r-2)^2s}{r-1}(f(x_k) - f^\star) + (r-1)\|z_k - x^\star\|^2,$$

where $z_k = (k+r-1)y_k/(r-1) - kx_k/(r-1)$. Suppose we have

$$\mathcal{E}(k) + \frac{2s[(r-3)(k+r-2)+1]}{r-1}(f(x_{k-1}) - f^\star) \leq \mathcal{E}(k-1). \tag{5}$$

Then it immediately yields the desired results by summing over (5). To be specific, by recursively applying (5) we see

$$\mathcal{E}(k) + \sum_{i=1}^k \frac{2s[(r-3)(i+r-2)+1]}{r-1}(f(x_{i-1}) - f^\star) \leq \mathcal{E}(0) = \frac{2(r-2)^2s}{r-1}(f(x_0) - f^\star) + (r-1)\|x_0 - x^\star\|^2,$$

which is equivalent to

$$\mathcal{E}(k) + \sum_{i=1}^{k-1} \frac{2s[(r-3)(i+r-1)+1]}{r-1}(f(x_i) - f^\star) \leq (r-1)\|x_0 - x^\star\|^2. \tag{6}$$

Noting that the LHS of (6) is lower bounded by $2s(k+r-2)^2(f(x_k)-f^\star)/(r-1)$ gives the first desired inequality. With $\mathcal{E}(k) \geq 0$, the second one is obtained via taking the limit $k \to \infty$ in (6) and replacing $(r-3)(i+r-1)+1$ by $(r-3)(i+r-1)$.

To complete, we aims to establish (5) in the rest of the proof. For $s \leq 1/L$ it is well-known in proximal gradient literature, for example [1], that

$$f(y - sG_s(y)) \leq f(x) + G_s(y)^T(y-x) - \frac{s}{2}\|G_s(y)\|^2 \tag{7}$$

for any $x$ and $y$. Note that $y_{k-1} - sG_s(y_{k-1})$ actually coincides with $x_k$. Summing of $(k-1)/(k+r-2) \times$ (7) with $x = x_{k-1}, y = y_{k-1}$ and $(r-1)/(k+r-2) \times$ (7) with $x = x^\star, y = y_{k-1}$ gives

$$
\begin{aligned}
f(x_k) &\leq \frac{k-1}{k+r-2} f(x_{k-1}) + \frac{r-1}{k+r-2} f^\star \\
&\quad + \frac{r-1}{k+r-2} G_s(y_{k-1})^T \left( \frac{k+r-2}{r-1} y_{k-1} - \frac{k-1}{r-1} x_{k-1} - x^\star \right) - \frac{s}{2}\|G_s(y_{k-1})\|^2 \\
&= \frac{k-1}{k+r-2} f(x_{k-1}) + \frac{r-1}{k+r-2} f^\star + \frac{(r-1)^2}{2s(k+r-2)^2} \left( \|z_{k-1} - x^\star\|^2 - \|z_k - x^\star\|^2 \right),
\end{aligned}
$$

where we use $z_{k-1} - s(k+r-2)G_s(y_{k-1})/(r-1) = z_k$. Rearranging the above inequality with multiplying by $2s(k+r-2)^2/(r-1)$ gives the desired (5).

$\square$

## 4 Proof of Theorem 5.2

**Remark 4.1.** *Indeed the linear convergence of $X^{\mathrm{sr}}$ remains for generalized ODE (4.1) with $r > 3$. Only minor modifications in proof such as replacing $u^3$ with $u^r$ in the definition of $I(t)$ in Lemma 4.1 are required to get analogous convergence rate for the speed restarting version of (4.1).*

**Lemma 4.1.** *The speed restarting time $T$ obeys*

$$T(x_0, f) \geq \frac{4}{5\sqrt{L}}.$$

*Proof.* Denote by $M(t)$ the supremum of $\|\dot{X}(u)\|/u$ over $u \in (0, t]$ and

$$I(t) \triangleq \int_0^t u^3(\nabla f(X(u)) - \nabla f(x_0))du.$$

By the proof of Lemma 1.5 it is guaranteed that $M$ defined above is finite. $M$ is useful in that it gives a bound on the gradient of $f$:

$$\|\nabla f(X(t)) - \nabla f(x_0)\| \leq L\|X(t) - x_0\| = L\left\| \int_0^t \dot{X}(u)du \right\| \leq L \int_0^t u\frac{\|\dot{X}(u)\|}{u} du \leq \frac{LM(t)t^2}{2}. \tag{8}$$

By (8), it is easy to see that $I$ can also be bounded via $M$:

$$\|I(t)\| \leq \int_0^t u^3\|\nabla f(X(u)) - \nabla f(x_0)\|du \leq \int_0^t \frac{LM(u)u^5}{2}du \leq \frac{LM(t)t^6}{12}. \tag{9}$$

To fully facilitate these bounds, we need to bound $M$ as

$$M(t) \leq \frac{\|\nabla f(x_0)\|}{4(1 - Lt^2/12)} \tag{10}$$

for any $t < \sqrt{12/L}$.

To this end, note that indeed ODE (1.2) is equivalent to $d(t^3\dot{X}(t))/dt = -t^3\nabla f(X(t))$, which by integration leads to

$$t^3\dot{X}(t) = -\frac{t^4}{4}\nabla f(x_0) - \int_0^t u^3(\nabla f(X(u)) - \nabla f(x_0))du = -\frac{t^4}{4}\nabla f(x_0) - I(t). \tag{11}$$

Dividing (11) by $t^4$ and applying (9), we obtain

$$\frac{\|\dot{X}(t)\|}{t} \leq \frac{\|\nabla f(x_0)\|}{4} + \frac{\|I(t)\|}{t^4} \leq \frac{\|\nabla f(x_0)\|}{4} + \frac{LM(t)t^2}{12}.$$

Note that the RHS of the above is monotonically increasing in $t$. Hence by taking the supremum of the LHS over $(0, t]$ we obtain

$$M(t) \leq \frac{\|\nabla f(x_0)\|}{4} + \frac{LM(t)t^2}{12},$$

which gives the desired (10) by rearranging the inequality for $t < \sqrt{12/L}$.

Having established (10), we proceed to lower bound $T$ via studying $\langle \dot{X}(t), \ddot{X}(t) \rangle$. Dividing (11) by $t^3$, one has an expression for $\dot{X}$, which reads

$$\dot{X}(t) = -\frac{t}{4}\nabla f(x_0) - \frac{1}{t^3}\int_0^t u^3(\nabla f(X(u)) - \nabla f(x_0))du. \tag{12}$$

Differentiating the above, we also obtain an expression for $\ddot{X}$:

$$\ddot{X}(t) = -\nabla f(X(t)) + \frac{3}{4}\nabla f(x_0) + \frac{3}{t^4}\int_0^t u^3(\nabla f(X(u)) - \nabla f(x_0))du. \tag{13}$$

Using the two expressions for $\dot{X}$ and $\ddot{X}$ we will show that $d\|\dot{X}\|^2/dt = 2\langle \dot{X}(t), \ddot{X}(t) \rangle > 0$ for $0 < t < 4/(5\sqrt{L})$. To this end, noting that (12) and (13) yield

$$
\begin{aligned}
\langle \dot{X}(t), \ddot{X}(t) \rangle =& \left\langle -\frac{t}{4}\nabla f(x_0) - \frac{1}{t^3}I(t), \ -\nabla f(X(t)) + \frac{3}{4}\nabla f(x_0) + \frac{3}{t^4}I(t) \right\rangle \\
\geq& \frac{t}{4}\langle \nabla f(x_0), \nabla f(X(t)) \rangle - \frac{3t}{16}\|\nabla f(x_0)\|^2 - \frac{1}{t^3}\|I(t)\|\left(\|\nabla f(X(t))\| + \frac{3}{2}\|\nabla f(x_0)\|\right) - \frac{3}{t^7}\|I(t)\|^2 \\
\geq& \frac{t}{4}\|\nabla f(x_0)\|^2 - \frac{t}{4}\|\nabla f(x_0)\|\|\nabla f(X(t)) - \nabla f(x_0)\| - \frac{3t}{16}\|\nabla f(x_0)\|^2 \\
& - \frac{LM(t)t^3}{12}\left(\|\nabla f(X(t)) - \nabla f(x_0)\| + \frac{5}{2}\|\nabla f(x_0)\|\right) - \frac{L^2M(t)^2t^5}{48} \\
\geq& \frac{t}{16}\|\nabla f(x_0)\|^2 - \frac{LM(t)t^3\|\nabla f(x_0)\|}{8} - \frac{LM(t)t^3}{12}\left(\frac{LM(t)t^2}{2} + \frac{5}{2}\|\nabla f(x_0)\|\right) - \frac{L^2M(t)^2t^5}{48} \\
=& \frac{t}{16}\|\nabla f(x_0)\|^2 - \frac{LM(t)t^3}{3}\|\nabla f(x_0)\| - \frac{L^2M(t)^2t^5}{16},
\end{aligned}
$$

where we use (9) and (8). To complete the proof, applying (10) in the above inequality yields

$$\langle \dot{X}(t), \ddot{X}(t) \rangle \geq \left(\frac{1}{16} - \frac{Lt^2}{12(1 - Lt^2/12)} - \frac{L^2t^4}{256(1 - Lt^2/12)^2}\right)\|\nabla f(x_0)\|^2 t \geq 0$$

for $t < \min\{\sqrt{12/L}, 4/(5\sqrt{L})\} = 4/(5\sqrt{L})$, where the positiveness follows from the fact that

$$\frac{1}{16} - \frac{Lt^2}{12(1 - Lt^2/12)} - \frac{L^2t^4}{256(1 - Lt^2/12)^2} > 0$$

for $0 < t \leq 4/(5\sqrt{L})$. $\qquad\square$

Next we give a lemma which claims that the objective function decays by a constant through each speed restarting.

**Lemma 4.2.** *There is a universal constant $C > 0$ such that*

$$f(X(T)) - f(x^\star) \leq \left(1 - \frac{C\mu}{L}\right)(f(x_0) - f(x^\star)).$$

*Proof.* By (11), (9) and (10) in Lemma 4.1, for $t < \sqrt{12/L}$ one has

$$\|\dot{X}(t) + \frac{t}{4}\nabla f(x_0)\| = \frac{1}{t^3}\|I(t)\| \leq \frac{LM(t)t^3}{12} \leq \frac{L\|\nabla f(x_0)\|t^3}{48(1 - Lt^2/12)},$$

which gives

$$0 \leq \frac{t}{4}\|\nabla f(x_0)\| - \frac{L\|\nabla f(x_0)\|t^3}{48(1 - Lt^2/12)} \leq \|\dot{X}(t)\| \leq \frac{t}{4}\|\nabla f(x_0)\| + \frac{L\|\nabla f(x_0)\|t^3}{48(1 - Lt^2/12)} \tag{14}$$

for $t < \sqrt{12/L}$. By Lemma 4.1 $d\|\dot{X}\|^2/dt \geq 0$ for $0 < t < 4/(5\sqrt{L})$ because $T \geq 4/(5\sqrt{L})$. Hence for $0 < t < 4/(5\sqrt{L})$ it yields that

$$\frac{df(X(t))}{dt} = -\frac{3}{t}\|\dot{X}\|^2 - \frac{1}{2}\frac{d}{dt}\|\dot{X}\|^2 \leq -\frac{3}{t}\|\dot{X}\|^2 \leq -\frac{3}{t}\left(\frac{t}{4}\|\nabla f(x_0)\| - \frac{L\|\nabla f(x_0)\|t^3}{48(1 - Lt^2/12)}\right)^2 \leq -ct\|\nabla f(x_0)\|^2,$$

where $c > 0$ is an absolute constant and the second last inequality follows from (14). Therefore we have

$$f(X\left(\frac{4}{5\sqrt{L}}\right)) - f(x_0) \leq \int_0^{\frac{4}{5\sqrt{L}}} -cu\|\nabla f(x_0)\|^2 du$$
$$= -\frac{c'}{L}\|\nabla f(x_0)\|^2 \leq -\frac{2c'\mu}{L}(f(x_0) - f^\star),$$

where the last step follows from the $\mu$–strong convexity of $f$. Above $c' > 0$ is an absolute constant. Thus we have

$$f(X\left(\frac{4}{5\sqrt{L}}\right)) - f^\star \leq \left(1 - \frac{2c'\mu}{L}\right)(f(x_0) - f^\star).$$

Last, recall that $f(X(t))$ decreases on $(4/(5\sqrt{L}), T)$, which finishes the proof by noting

$$f(X(T)) - f^\star \leq f(X\left(\frac{4}{5\sqrt{L}}\right)) - f^\star \leq \left(1 - \frac{2c'\mu}{L}\right)(f(x_0) - f^\star).$$

$\square$

To establish the linear convergence, we also need to ensure that $T$ can not be too large. To this end, we give the following lemma.

**Lemma 4.3.** *The speed restarting time $T$ satisfies*

$$T \leq \frac{4}{5\sqrt{L}}\exp\frac{C'L}{\mu}.$$

*Proof.* For $4/(5\sqrt{L}) \leq t \leq T$, we have

$$\frac{df(X(t))}{dt} \leq -\frac{3}{t}\|\dot{X}(t)\|^2 \leq -\frac{3}{t}\|\dot{X}(4/(5\sqrt{L}))\|^2,$$

which implies

$$f(X(T)) - f(x_0) \leq f(X(T)) - f(X(4/(5\sqrt{L}))) \leq -\int_{\frac{4}{5\sqrt{L}}}^{T} \frac{3}{t}\|\dot{X}(4/(5\sqrt{L}))\|^2 dt$$
$$= -3\|\dot{X}(4/(5\sqrt{L}))\|^2 \log\frac{5T\sqrt{L}}{4}.$$

Hence we get an upper bound for $T$ which reads

$$T \leq \frac{4}{5\sqrt{L}}\exp\left(\frac{f(x_0) - f(X(T))}{3\|\dot{X}(4/(5\sqrt{L}))\|^2}\right) \leq \frac{4}{5\sqrt{L}}\exp\left(\frac{f(x_0) - f^\star}{3\|\dot{X}(4/(5\sqrt{L}))\|^2}\right). \tag{15}$$

Plugging $t = 4/(5\sqrt{L})$ in (14) gives

$$\|\dot{X}(4/(5\sqrt{L}))\| \geq \frac{c}{\sqrt{L}}\|\nabla f(x_0)\| \tag{16}$$

for some universal constant $c > 0$. Substituting (16) in (15) yields

$$T \leq \frac{4}{5\sqrt{L}} \exp\left(\frac{L(f(x_0) - f^\star)}{3c^2\|\nabla f(x_0)\|^2}\right) \leq \frac{4}{5\sqrt{L}} \exp\frac{L}{6c^2\mu}.$$

$\square$

It readily gives the proof of Theorem 5.2 by combining Lemmas 4.2 and 4.3.

*Proof of Theorem 5.2.* According to Lemma 4.3, by time $t$ there are at least $n^\star \triangleq \lfloor 5t\sqrt{L}e^{-C'L/\mu}/4 \rfloor$ restartings for $X^{\mathrm{sr}}$. By Lemma 4.2 and monotonically decreasing of $f$ before restarting, we have

$$
\begin{aligned}
f(X^{\mathrm{sr}}(t)) - f(x^\star) &\leq f(X^{\mathrm{sr}}(\sum_{i=1}^{n^\star} T_i)) - f(x^\star) \\
&\leq (1 - \frac{C\mu}{L})(f(X^{\mathrm{sr}}(\sum_{i=1}^{n^\star-1} T_i)) - f(x^\star)) \\
&\leq \cdots \\
&\leq (1 - \frac{C\mu}{L})^{n^\star}(f(x_0) - f(x^\star)) \\
&\leq \exp(-\frac{C\mu n^\star}{L})(f(x_0) - f(x^\star)) \\
&\leq c_1(f(x_0) - f(x^\star))e^{-c_2 t\sqrt{L}},
\end{aligned}
$$

where $c_1 = \exp(C\mu/L)$ and $c_2 = 5C\mu e^{-C'\mu/L}/(4L)$. $\square$

## References

[1] N. Parikh and S. Boyd. Proximal algorithms. In *Foundations and Trends in Optimization*, volume 1, pages 123–231. 2013.