[Reviews · NeurIPS 2014]

Submitted by Assigned_Reviewer_17

This paper provides a theoretical and experimental analysis of Nesterov-type acceleration schemes for non-smooth, convex optimization. This is an important and widely-used class of algorithms, and in recent years they have gathered a tremendous amount of interest from numerous researchers in machine learning. Nesterov's algorithm, FISTA and the numerous variants they have spawned represent cornerstones on which many large-scale optimization algorithms rely.

The paper begins from the observation that the iterates of Nesterov-type algorithms behave like a discretized version of a damped oscillator. While this is almost certainly a known fact (it is clear after implementing such a scheme and observing the iterates), one of the main contributions of this paper is to go the extra step and make this intuition precise. Specifically, the author(s) derive a second order ordinary differential equation and demonstrate that Nesterov's scheme can be viewed as a numerical integration procedure applied to this ODE. This analogy might also be known, or at least used implicitly in other works. The author(s) seem well aware of this possibility. This possibility does not damage the paper, as the author(s) go much further and actually use this insight in novel ways. In particular, the author(s) use the ODE analogy to derive several new algorithms in the Nesterov class. The paper then applies classical ODE analysis, in the form of energy estimates, to provide simple convergence proofs as well as temporal rates of convergence. The discrete versions of these proofs provide analogous convergence rates for the new algorithms.

Although it is not the main thrust of the paper, whether the proposed algorithms would yield practical improvement in applications is unclear. The asymptotic rate for the generalized schemes in theorem 4.2 (first display line) does not improve upon the rate of existing algorithms, for instance. Theorem 4.3 does provide an improved rate for this family of schemes under a strong convexity hypothesis, but there exist asymptotically better (linearly convergent) schemes under these same hypotheses. So-called ``generalized ADMM'' schemes are linearly convergent in this case, for instance. On the other hand, the subproblems in such ADMM schemes are typically more difficult to solve than those in the generalized Nesterov scheme. Finally, although the author(s) suggest a linearly convergent scheme via restarting, a proof is only provided at the ODE level. A proof of linear convergence for the discrete scheme, in some form, would certainly have made the paper stronger.

Overall this is a good paper. Due to the explosion in popularity of accelerated first-order methods, any work that offers deeper insight and a more unified approach to studying them is welcome. The author(s) suggest that a refinement of the ODE analogy could provide such a framework. The paper might have been stronger had this been explored further, such as using ODEs to model accelerated primal-dual schemes or other accelerated first-order methods. Nevertheless, the present work is sound.
Summary: The paper provides a solid and insightful analysis of an important class of algorithms, but the practical improvements offered by the proposed algorithms are unclear.

Submitted by Assigned_Reviewer_41

The authors present a novel (or classical, depending on the perspective) take on Nesterov's accelerated gradient schemes. They investigate the continuous analogue of the update rule in Nesterov's momentum-based scheme, and then study the asymptotic dynamics of this analog. In particular, they argue that the O(1/t^2) convergence rate of Nesterov's scheme on smooth functions follows in a straightforward in the asymptotic version, and can be obtained for a slightly more generalized version of the update. They also propose a composite version and analyze continuation schemes in the strongly convex setting.

I believe that the take of this paper on Nesterov's method is certainly of interest. While asymptotic analyses using ODE methods have been classical in optimization theory, the treatment of some of the more modern methods has been largely focused on understanding the finite sample properties. The two approaches typically have complementary strengths---asymptotics usually yield sharper (problem dependent) constants and are often easier to establish in broader settings, while finite sample analysis yields insights into how long it takes before the asymptotics can kick in and the properties of the algorithm affecting this initial period. From that point of view, having a clear understanding of the asymptotics of Nesterov's scheme is certainly valuable.

At the same time, I am bothered by certain technical sloppiness and errors which I am unable to verify or correct, leaving me unable to assess the correctness of the result. I will describe the problems below. If the authors can convince me of the correctness, then I am happy to recommend acceptance of this work, but without it the paper falls short of the standard for me.

The first couple of technical issues arise in obtaining the dynamics (2) from (3). Specifically, I am unable to verify the expression relating \nabla f(y_k) to \nabla f(X(t)) on line 86, and could not find an explanation. It seems to require that x_k - x_{t-1} = o(1), which does not seem correct in this constant stepsize setting.

Secondly, the authors replace the (k-1)/(k+2) term with 1 - 3/k in Equation 4. This seems problematic, since the error would be lower order in isolation but it is also being scaled with (x_k - x_{k-1}). In particular, it is not clear if the error, multiplied with the \ddot{X(t)}\sqrt{s} term is still of the correct order. I understand that the authors would like to reason with asymptotics, but it is still important to reason about the order of all the error terms, and I believe the treatment here seems somewhat imprecise.

The other technical problem seems to arise in the proof of Theorem 3.2 as well as 4.1. It seems like the derivative of the energy functional in Equation 7 should have \dot{X} + t\ddot{X} instead of 3\dot{X} + t\ddot{X} in the last term. Similarly, r\dot{X} in Equation 9 should be replaced with \dot{X}/(r-1) it seems. Changing these constants seems to break the proof, since you can no longer replace the term in the inner problem with a rescaled version of \nabla f(X), and that too in a way which will precisely cancel the other term. Maybe I made an error in differentiation, but I am unable to see why the proofs are correct.

In terms of the continuation scheme, the authors seem to ignore that Nesterov indeed analyzes a continuation method adaptive to the strong convexity constant in his paper "Gradient methods for minimizing composite objective functions". Section 5.3 of that paper discusses an adaptive method which only loses additional logarithmic factors compared to the case with a known strong convexity constant. This is a much stronger result than Theorem 5.2, which seems to leave the dependence of c_1 and c_2 on the strong convexity constants open, despite being in a simpler, asymptotic setting.

Overall, I find that the paper cannot be accepted as it is with the technical issues. If the authors can convince that this is not a problem in their rebuttal, I will reconsider my decision.
Summary: The authors present an ODE to capture the aysmptotics of Nesterov's accelerated methods, and show that the rates of convergence of accelerated methods on smooth functions can be easily obtained by analyzing the ODE. However, there seem to be technical flaws in the paper which make it unacceptable for publication in the current form.

Submitted by Assigned_Reviewer_43

==========added after author rebuttal============
Re Theorem 4.3. I think my initial comment is correct but maybe I should have been more explicit: Eq. (36) in Tseng's paper or the similar stepsize in Beck and Teboulle need not be satisfied with equality. When one bounds the suboptimality wrt a global minimizer, such as the f* in the author's Theorem 4.3, the inequality <= in Tseng's Eq. (36) is enough, which makes the stepsize (r-1)/(k+r-1) feasible. The strict equality is needed when one compares wrt an arbitrary point, such as in the theorem Tseng derived. This is apparent if one checks when Eq. (36) is used in the proof, or when Beck and Teboulle applied their stepsize rule (proof of their Lemma 4.1, 2nd displayed equation in page 195, note that v_k >=0).

Re Experiments: The authors seem to misunderstand my comment. I was not asking for a thorough comparison with a great number of competitive algorithms, but a few that naturally pop out in one's mind. These comparisons not only give us a better picture but also provide guidance for others who might be interested in applying the proposed algorithms.
======================================

Nesterov's accelerated gradient algorithm has been a very popular algorithm for handling large datasets, and is known to be optimal among first order methods. The authors re-derived Nestorov's algorithm as a discretization of a second order differentiable equation. It appeared the convergence of the continuous ODE is much cleaner to deal with. Moreover, inspired by the continuous ODE, the authors proposed two variations, one of which could exploit strong convexity even when the parameter is not known beforehand. Limited experimental results partially validate the theoretical discoveries.

As far as the reviewer is concerned, the convergence results derived in this submission are novel and potentially significant. Although the idea to relate continuous ODE with discrete gradient methods has a long history, the authors have done a superb job in instantiating the general principle to Nesterov's algorithm, overcoming some technical difficulties along the way.

Theorem 4.3 is not quite as new as it appears to be. In fact, the update rule in Eq. (12) makes it clear that it is nothing but the same update rule in Eq (34) in the following paper:
"Approximation accuracy, gradient methods, and error bound for structured convex optimization, Tseng'10", under the identification theta_k = (r-1) / (k+r-1). In fact, Tseng's treatment is even (slightly) more general, with a very enjoyable proof too. This should be pointed out. The equation in line 260 seems to be new and interesting.

Theorem 4.4 is very interesting as it does not require the knowledge of the strong convexity parameter. Unfortunately the current proof appears to be long and tedious. It would be great to polish the proof to further distill the essence. On the other hand, in many machine learning applications the strong convexity indeed is known beforehand (in particular if one uses strongly convex regularizers).

The current experiments are not very satisfying. Two very natural comparisons are missing (when strong convexity is present): 1). the usual proximal gradient (without accelerating), which achieves a linear rate with a worse constant; 2). Nesterov's algorithm with the knowledge of the strong convexity parameter. These comparisons will give us a better idea on how effective the restarting strategy is. The experiment suffers also from another apparent issue: Nesterov's original algorithm's oscillating behavior. It is trivial to fix this issue by enforcing monotonicity, i.e., picking y_t as the best (in terms of objective value) of y_t and y_{t-1}. The convergence property will not he jeopardized (because the momentum term is still changing). This is a third competitor the authors should have compared to.

Although the authors did give some justification of the restarting strategy, the results are not as convincing as it could be at the moment: 1). Theorem 5.2 is about the continuous ODE and does not allow any nonsmooth component; 2). How does it compare to a simple halving trick, i.e., one starts with a guess mu of the strong convexity parameter and keeps halving it?
Summary: This work provides a fresh ordinary differential equation (ODE) viewpoint of Nesterov's accelerate gradient algorithm. The continuous ODE is much simpler to deal with and gives insight to build other variations, continuous or discrete. One particular restarting strategy is shown to work well for strongly convex functions even when the strong convexity parameter is not known or zero. The results seem to be interesting and novel.
Author Feedback
Author rebuttal: We would like to thank the referees for the time they spent reviewing our paper. We have found their comments to be thoughtful. In this letter, we wish to respond to the issues that were raised.

Reviewer 17 states that: “Specifically, the author(s) derive a second order ordinary differential equation and demonstrate that Nesterov's scheme can be viewed as a numerical integration procedure applied to this ODE. This analogy might also be known, or at least used implicitly in other works. The author(s) seem well aware of this possibility. This possibility does not damage the paper, as the author(s) go much further and actually use this insight in novel ways.”

While it is true that there is a tight connection between ordinary differential equations and optimization schemes, we were not able to find the analog of the ODE (2) in the literature. In particular, we could not find the main observation we make, which is that the time parameter in a Nesterov scheme scales like root of the step size.

In addition, Reviewer 17 comments that: “Although it is not the thrust of the paper, whether the proposed algorithms would yield practical improvement in applications is unclear. The asymptotic rate for the generalized schemes in theorem 4.2 (4.3?) (first display line) does not improve upon the rate of existing algorithms, for instance.”

In terms of the complexity of smooth minimization, we can not expect a better rate than 1/k^2, which is achieved by the generalized Nesterov’s schemes proposed in Section 4.2. On the other hand, in Theorem 4.3 the second display, which bounds the errors in an average sense, seems to be new in the literature. Given the popularity of Nesterov’s method and its variants in modern large-scale optimization, we expect that this class of generalized Nesterov’s schemes has the potential to find some practical applications. Theorem 4.4 itself intend to characterize the newly proposed schemes in a different way, rather than advertising these schemes for minimizing strongly convex functions.

Reviewer 41 reports some issues regarding technicalities, including: “The first couple of technical issues arise in obtaining the dynamics (2) from (3)”; “Secondly, the authors replace the (k­1)/(k+2) term with 1 ­ 3/k in Equation 4”; “The other technical problem seems to arise in the proof of Theorem 3.2 as well as 4.1”. We clarify these issues in the following three paragraphs.

In line 86 we have used the approximation y_k - X(t) = o(1). This approximation is valid because the step size s is assumed to be vanishing. Specifically, y_k - X(t) \approx y_k - x_k = (k-1)/(k+2)(x_k - x_{k-1}) = o(s).

As for the second point, the difference between (k-1)/(k+2) and 1 - 3/k is O(1/k^2). Note that in our ansatz k = t/\sqrt{s}, which leads to O(1/k^2) = O(s). This O(s) term is absorbed in o(\sqrt{s}) in line 92. If possible, we are willing to make these approximations more precise.

The proofs of Theorems 3.2 and 4.1 are correct. The derivative of the energy functional in (7) is indeed 3\dot{X} + t\ddot{X} in the last term because the derivative of t\dot{X} is \dot{X} + t\ddot{X}. Analogously, the correct term in (9) is r\dot{X} rather than \dot{X}/(r-1).

Reviewer 41 further states that: “In terms of the continuation scheme, the authors seem to ignore that Nesterov indeed analyzes a continuation method adaptive to the strong convexity constant in his paper ‘Gradient methods for minimizing composite objective functions’.”

We are aware of some algorithms achieving linear convergence for strongly convex functions by adapting to unknown parameters. Compared to the literature, the linear convergence stated in Theorem 5.2 is obtained by simple restartings. This is also pointed out by Reviewer 41.

Reviewer 43 comments that: “Theorem 4.3 is not quite as new as it appears to be. In fact, the update rule in Eq. (12) makes it clear that it is nothing but the same update rule in Eq (34) in the following paper:"Approximation accuracy, gradient methods, and error bound for structured convex optimization, Tseng'10", under the identification theta_k = (r­-1) / (k+r-­1).”

The updates (34), (35) and (36) in Tseng’s paper are first proposed in the paper “A fast iterative shrinkage-thresholding algorithm for linear inverse problems” by Beck and Teboulle, which is a reference of our paper. In fact, the update (36) excludes the possibility that \theta_k = (r­-1) / (k+r­-1). Specifically, it can be checked that with any starting point \theta_0, iterating according to (36), \theta_k = 2/k + O(1/k^2) asymptotically.

Reviewer 43 also states that: “The current experiments are not very satisfying” and “Although the authors did give some justification of the restarting strategy, the results are not as convincing as it could be at the moment”.

In understanding of Nesterov’s schemes along with providing some refinements and variations, our goal is not to compare our schemes with a great number of other first-order methods. However, if possible, we are willing to entertain a limited comparison with some widely used first-order methods.